# Grading by size to reduce the opportunity for domestication selection in hatchery-reared steelhead (*Oncorhynchus mykiss*)

Miriam B. Obley[1,¤a,*], Ruth H. Milston-Clements[2,¤b], Stephanie R. Bollmann[1],
Michael S. Blouin[1]

1 Department of Integrative Biology, Oregon State University, Corvallis, Oregon, United States of America,
2 Department of Microbiology, Oregon State University, Corvallis, Oregon, United States of America

¤a Current address: Department of Fisheries, Wildlife, and Conservation Sciences, Oregon State University, Corvallis, OR, USA
¤b Current address: Division of Research and Innovation, Oregon State University, Corvallis, OR, USA
* obleym@oregonstate.edu

## Abstract

Fish that are produced in hatcheries often produce fewer surviving adult offspring than do wild fish when both spawn in the wild. This difference in fitness is likely due to inadvertent selection for adaptation to the hatchery environment. Size at release is positively correlated with survival at sea. Therefore, selection should favor traits that promote fast growth in the hatchery even if those traits are maladaptive in the wild. In that case, changing hatchery conditions to reduce the variance in size at release would reduce the opportunity for domestication selection. Here we test whether grading by size and raising each size group separately can substantially reduce the variance in size at release. We graded a mix of 15 full-sibling families of winter run steelhead (*Oncorhynchus mykiss*) juveniles (initial n = 375 fish/tank) into small, medium, and large body size groups. We then promoted growth in the initially-small fish with excess food and low densities, and slowed growth in the initially-large fish with restricted feedings and high densities. We successfully held back the growth of the initially-large fish, but the initially-small fish never caught up with the controls, despite being raised under ideal conditions. This result suggests that inherent physiological or behavioral factors, rather than interactions with larger fish, restrict the growth of small fish. Both the total variance among individual fish and the variance among family means was reduced by the grading treatment, although only the reduction in total variance among individuals was statistically significant when comparing intraclass correlation values (ICC). The reduced variance due to slowing the growth of the initially-large fish indicates that grading could possibly be a tool used by hatcheries to lessen selection on salmonids. However, it would first be necessary to better understand the factors limiting growth in the smaller fish.

**Data availability statement:** All data generated or analyzed during this study are available in the following public repository: https://github.com/obleym/Grading_Data_Obley.

**Funding:** This research was supported by the Oregon Hatchery Research Center (https://www.dfw.state.or.us/fish/ohrc/). The funders had no role in study design, data collection and analysis, decision to publish, or preparation of the manuscript.

**Competing interests:** The authors have declared that no competing interests exist.

## Introduction

Hatchery programs have been heavily implemented in the Pacific Northwest to mitigate the decline of wild salmonid populations. As a valuable resource culturally, ecologically, and economically [1–3], hatcheries are dedicated to maintaining and preserving salmon availability for food, angling, and can be used for conservation [4,5]. While programs like these are successful at producing fish for harvest, there is less evidence supporting supplementation programs on the recovery of wild populations and the success of their progeny [6,7]. In contrast, fish reared under hatchery conditions tend to have lower reproductive success when spawning in the wild than natural origin fish [8,9]. This decrease in fitness is likely due to domestication (i.e., adaptation to the hatchery setting), which can lead to behavioral and behavioral and physiological differences between captive and wild populations [10–12] and may be responsible for the epigenetic differences between hatchery and natural origin salmonids [13]. However, there is a tradeoff that occurs in which the well adapted fish in the hatchery do poorly in the wild and vice versa [14]. Increased growth rates in captivity have been observed in the offspring of captive-reared fish compared to the offspring of natural origin fish [15–18]. Growth in the hatchery has also been linked to survival at sea because larger fish that are larger at release are more likely to return than smaller fish at release [19–21].Therefore, any trait variation among fish in a hatchery that influences their growth in this novel environment may be under strong selection. Furthermore, there is substantial variation in size at release among fish reared in captivity [22], family identity explains substantial variance in size among fish reared in hatcheries [15,23], and growth in hatcheries tends to have high heritability [24]. Hence, there is also substantial opportunity for response to selection on whatever traits are behind variation in growth rate under hatchery conditions.

So what traits might be under selection in hatcheries? Hatchery fish tend to be more aggressive than wild fish [25] indicating boldness or aggression might be selected for in the hatchery environment. It is well known that individual fish vary in a correlated suite of traits that includes boldness, aggressiveness and dominance [26,27]. These "pace of life" traits tend to be consistent among individuals, highly heritable, and also correlated with physiological traits such as metabolic rate [28,29]. Bolder, more aggressive fish gain more access to food, have a higher metabolic rate, and subsequently grow faster and more efficiently than subdominant fish reared in the same environment [30–32]. However, excess boldness is not necessarily beneficial in the wild where other behaviors such as predator avoidance may be important to tradeoffs between growth and survival [33,34].

If the wild-born offspring of hatchery fish inherit the tendency for bold/aggressive behaviors, then they might have low survival in the wild. Thus, inadvertent selection in the hatchery for heritable traits such as boldness/aggression could also explain the reduced fitness of hatchery fish when they spawn in the wild.

Aspects of fish physiology and behavior can be manipulated by changing the rearing environment [35]. Therefore it has been hypothesized that altering environmental conditions in the hatchery could reduce the variance in size at release, thereby reducing the opportunity for selection on whatever traits influence growth

in the hatchery [23]. In particular, if hatchery conditions typically favor bolder/aggressive phenotypes, then introducing conditions that favor the shyer fish might reduce the variance in size at release. However, simple changes to the hatchery environment designed to favor shyer phenotypes, such as reducing rearing density, increasing water flow, or adding physical structure to tanks have achieved little reduction in the variance of fork length [23,36]. In general, the families that grow quickly in one environment also grow quickly in others, and vice versa for the slow-growing families. More substantial changes may need to be made in order to reduce variance in size at release.

We hypothesize that grading fish based on size (fork length), and then rearing the different size groups separately, may be one method that is more effective than small environmental changes without adding too much complexity to hatchery procedures. Grading is used in hatchery productions to create better angling opportunities, but as a result, smaller fish are often discarded [37,38]. By separating large and small fish and keeping the small group, it may have the opportunity to gain access to more food, be relieved of aggressive interactions with larger (presumably bolder) fish and have the potential to grow larger and at a faster rate than they would under typical conditions [39,40]. In contrast the large group's growth rate may be slowed by feed restriction so that all of the fish in the entire cohort would be more likely to reach the same target length at the time of release. Thus, grading may be an effective method to overcome inherited traits among families of hatchery-reared salmonids and therefore reduce the opportunity for inadvertent domestication selection.

Alternatively, it may be that the slower growth rate of the small fish in a hatchery setting may not result from interaction with more aggressive fish, but from inherent physiological or behavioral traits that limit their ability to take advantage of the hatchery setting. There may be some trait that the small fish have that keep them from growing rapidly in the hatchery regardless of their interaction with the larger fish. In that case, separating out the slower growing fish would have little effect on their performance.

Here we test whether grading a cohort of steelhead into fast, medium and slow growing groups, and then deliberately manipulating the target release sizes of each group via food and density, can substantially reduce the overall variance in size of the entire cohort at release.

## Materials and methods

### Spawning and incubation

All fish handling protocols were approved by the Oregon State University Institutional Animal Care and Use Committee (Animal Care and Use Protocol IACUC-2021–0236). Winter-run steelhead were collected by Oregon Department of Fish and Wildlife (ODFW) staff at the Siletz Falls Trap on the Siletz River, Oregon, and spawned by ODFW staff at the Oregon Hatchery Research Center (OHRC) (coordinates 44.4040, −123.7532). Ten pairs were spawned on March 25, 2022, and six pairs on April 7, 2022 for a total of 16 individual families (A-P). Fin clips from all parents were collected and stored in 95% ethanol for future genotyping and sorting offspring into their families. All eggs were water hardened and treated in a thiamine bath for 1 hour as prophylaxis against thiamine deficiency. On both spawning dates, only one female was spawned with each male, creating 16 full sibling families that were raised in individual family trays in vertical stack incubators until they hatched and absorbed yolk sacs. The first spawn group was held on chilled water in order to sync the growth of the two groups. Fry were then transferred to separate Vexar mesh baskets in a trough with ambient temperature water flowing through. Food was withheld for 2–3 days to avoid fungal growth. Fry were then fed and remained in the trough until they learned to feed exogenously.

### Sorting into treatment tanks

On June 22, 2022, fish from each of the families were transported in coolers containing aerated river water to the John Fryer Aquatic Animal Health Lab (AAHL) (coordinates 44.5759, −123.2403) at Oregon State University (approximately 1 hour drive). Upon arrival, twenty-five fish from each of the 15 families were added to each of 4 experimental tanks (n = 375

fish/tank). Experimental tanks were 380 L, 3 ft diameter x 3 ft depth circular fiberglass tank with constant flow-through, UV sterilized 13 °C well water. Tanks were located outdoors and therefore received a natural photoperiod.

## Project design – grading

Fish were fed a diet of BioOregon Bio Vita starter feed crumble and then increasing sized crumble and pellets at a frequency, pellet size and % body weight (%BW) according to ODFW feed guidelines calculated to reach target weights (S1 file). Initially, a feed regime to reach a target release weight of 6 fish per pound (FPP) (~75g/fish) by April 14 2023 was chosen. This is in line with current hatchery practices for release size and timing of release of Winter Steelhead at the Alsea ODFW Hatchery [41].

After 143 days fish had grown sufficiently such that there was a range of fork lengths (FL) within each tank. Fish from two of the four tanks were graded by measuring FL of individual fish and splitting into three groups depending on FL: large (>113 mm FL), medium (97–113 mm FL), and small (97 mm FL). This resulted in 2 tanks (380 L capacity) for each FL size bin. To create a high-density environment as an additional way to limit growth of the large fish, the fish from the two large FL tanks were then pooled into one 380 L tank (Fig 1). Two of the four original ungraded tanks remained as control tanks.

In this experimental design our goal was to compare the variance in FL of the fish that were not graded with the variance in FL of the fish that were graded, and it is fish from the same 15 families in each treatment group. In other words, at the end of the experiment there will be just two groups of fish that will be compared. One group was raised under standard (control) conditions. The other group was divided into grading size groups that were treated differently, and then combined into one group at the end of the experiment. Note that we are not using individual tanks as units of replication of a treatment, but just to increase the number of fish we could raise under each condition.

At this stage, control and medium tanks continued to be fed based on % BW with the goal of reaching 6 FPP (~75g/fish) by April 14, 2023 according to ODFW feed guidelines chart (S1 File). To encourage smaller fish to grow more, small tanks were fed more, with a target release weight goal of 4 FPP (~113g/fish). Conversely, the large tank was fed less, with a goal of 7 fish per lb. The large tank was also fasted two days a week to limit growth. Our goal was to have the initially-small and the initially-large groups reach the same size as the initially-medium group at the end of the experiment. Fish were sub-sampled for length and weight to adjust feed rations at least once a month during this rearing period.

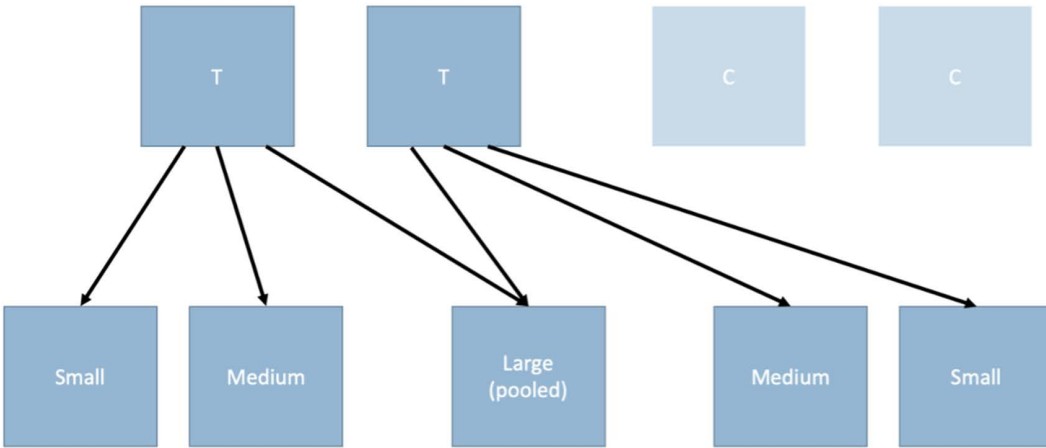

**Fig 1. Two treatment tanks graded into new tanks based on fork length.** T = treatment, C = control. Fish were moved into small, medium, or large tanks. The large fish were pooled into one tank to create a higher density environment. Two tanks were kept as controls. T = treatment, C = control.

Fish were euthanized on April 15, 2023 using MS222 (dose 10 ml buffered MS222/ 1Liter $H_2O$), and were measured for fork length. A fin clip from each individual fish taken and stored in 95% ethanol to preserve the DNA for genetic parentage analysis in order to be matched back to their individual families.

### Health monitoring and treatments

Fish were monitored daily for health status. From July 2, 2022 to July 11, 2022, all fish were given Florfenicol medicated feed at 2% of the body weight for 10 days to treat for Bacterial Cold Water Disease (Flavobacterium psychrophilum). From August 1, 2022 – August 3, 2022 fish were given daily formalin baths to treat for the parasite Ichthyobodo necator (Costia). These pathogens were present due to their initial rearing in natural river water system at the OHRC. Mortality was negligible and did not impact specific families disproportionately.

### Genetic parentage analysis

All samples were genotyped and assigned back into their original families via genetic parentage analysis. DNA was extracted from preserved fin clips with Chelex 100 [42]. Six microsatellite loci (Ogo4, Omm1046, Omy7, One102, Ots4, Ssa407MP) from the SPAN B suite were multiplexed and amplified with PCR [43] in a single reaction. All samples were submitted to the Center for Quantitative Life Sciences Core Facilities at Oregon State University where genotype scoring was performed on an AB 3730 capillary DNA sequencer. ABI GeneMapper® software was used to analyze the output (Applied Biosystems 3730 Genetic Analyzer, RRID:SCR_018052). Each sample was assigned back to its family using the parent genotypes and pure-exclusion parentage analysis, requiring unambiguous genotypes from each offspring for at least five loci.

### Statistical analysis: Opportunity for selection

At the end of the experiment, we considered all the fish from the grading tanks to be one cohort, and all the fish from the control tanks to be another cohort. Thus, we are comparing two groups of fish, and individual fish (not tanks) are the unit of replication. We examined the effect of grading on the total variance in size (FL) among fish, and on the among-family component of variance in size. We examined the among-family component of variance because family should be a reasonable proxy for heritable variation in size [24]. Before examining the effect of treatment on total variance among individuals and on variance among family means, we adjusted the fork length of each individual fish to remove the random tank effects in each treatment and size group. We did this because we are interested in the total variance of fish in each group, uninflated by random tank effects. We adjusted the two control tanks to have the same mean, and each group within the grading treatment to have the same mean (i.e., the small tanks had the same mean and the medium tanks had the same mean; there was only one large tank so length was not adjusted). By adjusting length this way, the random tank effects were accounted for without removing the effect of grading. Hereafter, we use the term "treatment" to refer to the control group versus the graded group (not to the small/medium/large feeding and density regimes that were applied to the graded group).

We compared the total variance in size, and the variance among family means, between the two treatments (graded vs. control) via Fligner Killeen median tests because of the left-hand skew in the distribution of the data (Fig 2). To further assess the effect of treatments on the among-family variation in length, we also calculated the intraclass correlation (ICC), which is the percentage of total variation in length that is distributed among families. A higher ICC value indicates more opportunity for selection among families. So, we are testing whether the treatment significantly reduced the ICC. The ICC package "ICCest" in R was used to calculate ICC values and confidence intervals [44].

## Results

At the end of the experiment, the total distribution of size within each treatment appeared close to normal but with a left-hand skew of very small individuals (Fig 2). Fish were 0.49 cm smaller overall in the graded cohort ($t = -5.18$, df = 1167, p

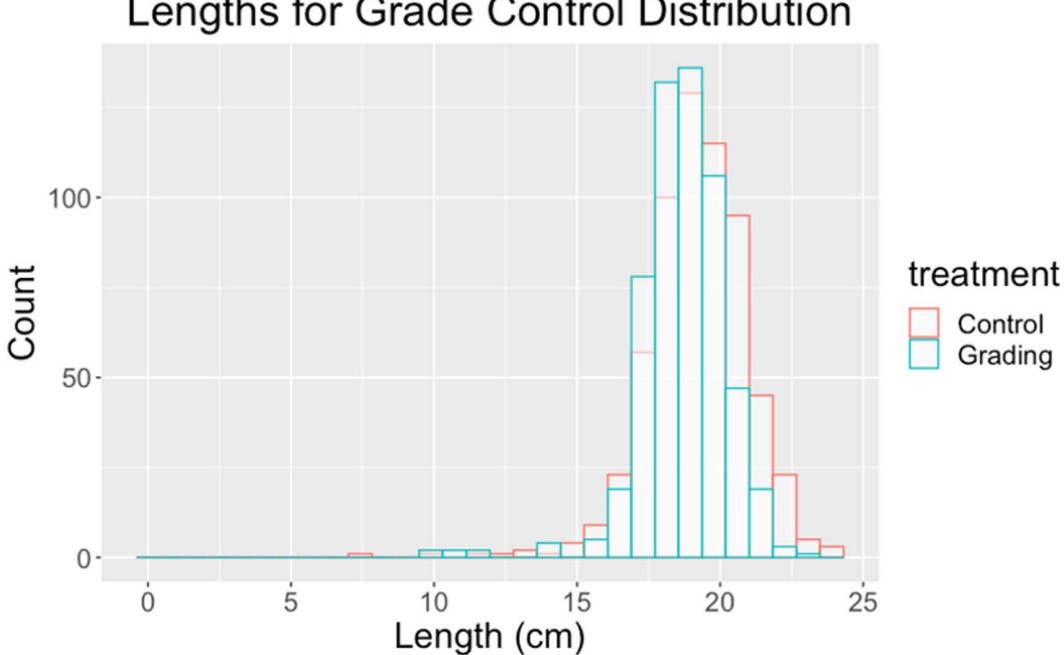

**Fig 2. Histograms of body length of all the fish in each treatment.** The size of each fish was adjusted to remove random tank effects within each treatment. Distributions were close to normal, but with a left-hand skew due to very small fish.

value = 2.61e-07). From Fig 2, it appears that this reduction in mean size results mainly from a reduction in the right-hand tail of the distribution relative to controls, with little change in the left-hand tail.

The origin of this pattern is obvious in Fig 3, which shows the average sizes of the fish in each feeding regime at the beginning and end of the experiment. The growth of the "large" grading group was successfully held back, such that at the end of the experiment they were smaller on average than the "medium" grading and control groups. However, fish in the "small" grading group never caught up, despite being raised at very low densities and being fed substantially more than the other groups (Fig 3).

We can examine the effect of the grading treatment on the overall variance in size, on the among-family variance in size, and on the sizes of individual families. As expected, families were unequally distributed throughout the small, medium, and large grading tanks at the start of the experiment (Fig 4). For example, there were very few members of families D, G, K, N, O, and no members of family M, represented in the initially-small grading tanks.

At the end of the experiment, the average sizes of these initially-larger families were substantially reduced in the graded cohort relative to in the control cohort (Fig 5). While the spread of family means in the grading treatment is reduced, most of this reduction appears to result from holding back the initially fast-growing families, rather than improving the growth of the initially-slow-growing families (Fig 5).

The total variance in body size was 25% smaller in the graded group than in the control group, a significant difference based on the Fligner Killeen median test (P = 7.65e-05, chi-squared = 15.64; control var = 2.96, graded var = 2.23, n1 = 615, n2 = 556). The variance among the 15 family means in the graded group (Var = 0.39) was 39% smaller than in the control group (Var = 0.64) group, although this difference was not statistically significant via the Fligner Killeen test (P = 0.076, chi-squared = 40.51; n1 = 15, n2 = 15). This difference in statistical significance is presumably owing to the small sample size of only 15 families in each group vs. n = 615 and 556 in the test using individual fish. Similarly, the percentage of the total variance that was distributed among families (ICC value) was smaller in the graded group (0.15) than in the control group (0.21), but with broadly overlapping 95% confidence values (Table 1 and Fig 6).

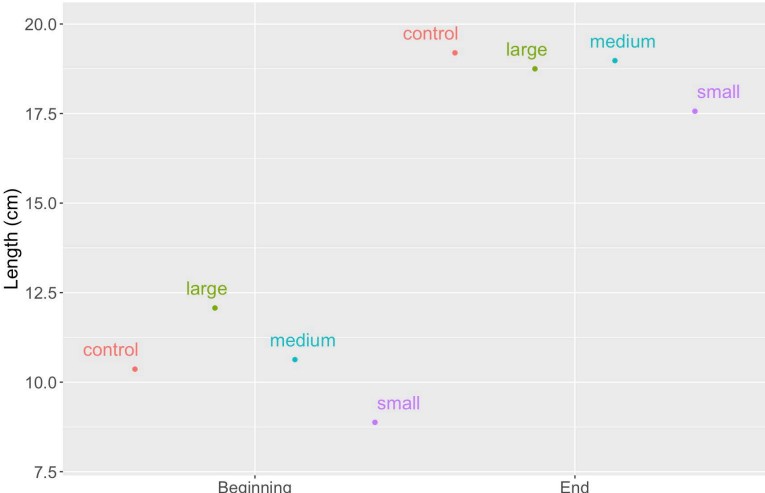

**Fig 3. Comparison of mean fork length in grading and control tanks at the beginning and end of the experiment.** At the end of the experiment, the initially-large grading group was smaller than the initially-medium grading and control groups. The initially-small grading group did not catch up to the other groups.

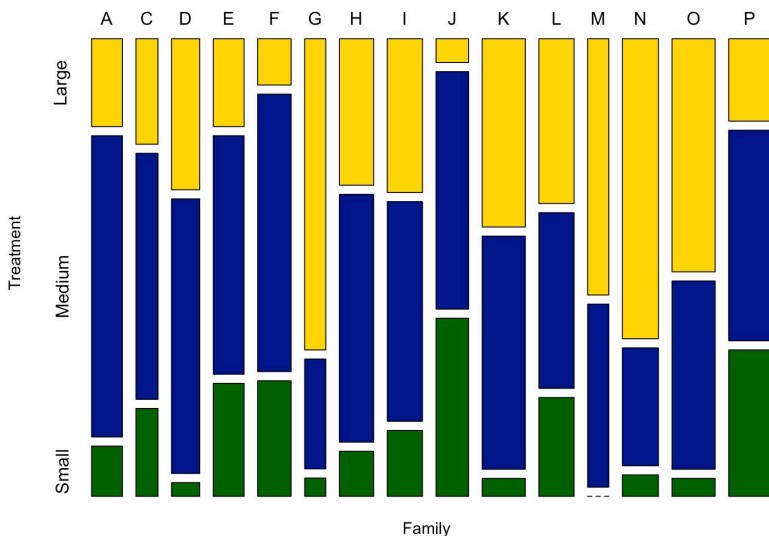

**Fig 4. Distribution of families throughout the grading tanks.** Families were unequally represented in each grading regime. Letters (columns) represent individual families, and the stacked bar charts show the percentage of each family that wound up in the initially-small (green), initially-medium (blue) and initially-large (yellow) groups.

Thus, the grading treatment did achieve a reduction in the nominal values of total variance in body size among all fish in the treatment, in the variance among family means (our proxy for heritable variation in size), and in the among-family component of the total variance, although only the reduction in variance among individual fish was statistically significant.

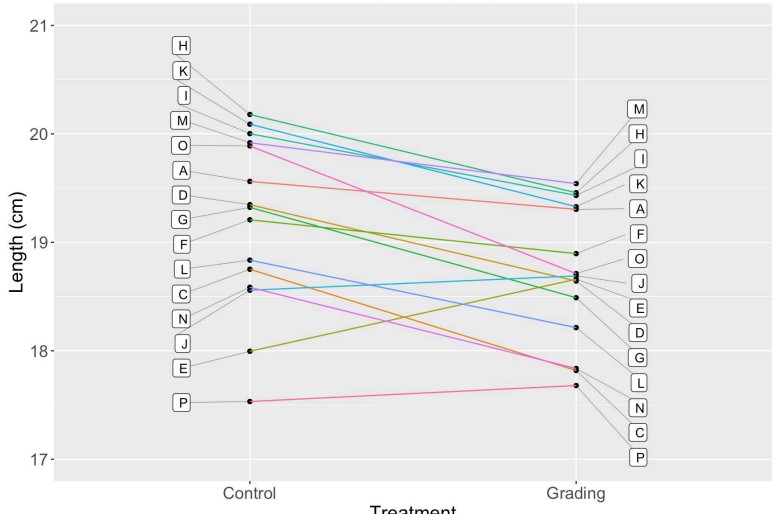

**Fig 5. Comparing family mean fork lengths between control and grading treatments.** The larger families had substantially reduced means in the grading treatment, which resulted from their overrepresentation in the initially-large treatment group, which received less food. The effect of the grading treatment on the initially-smaller families was not as apparent. The variance among family means is 39% lower in the graded cohort than in the control cohort. Letters identify families, and colored lines identify the same family across the two treatments.

**Table 1. Intraclass correlation (ICC) values and variance components for control and grading treatments.**

| Treatment | ICC | $V_w$ | $V_a$ | Lower CI | Upper CI |
|---|---|---|---|---|---|
| control | 0.21 | 2.37 | 0.63 | 0.073 | 0.35 |
| grading | 0.15 | 1.90 | 0.34 | 0.040 | 0.27 |

$V_w$ = component of variance within each family; $V_a$ = component of variation among families; CI = 95% confidence interval.

## Discussion

Grading fish into three groups and raising them with different growth targets produced a cohort that had a smaller total variance in size than the control cohort, which was reared under standard conditions. The variance among family means, and the percentage of total variance distributed among families, was also smaller in the graded cohort, although not significantly so. Nevertheless, a nominal 39% reduction in the variance among family mean size seems biologically interesting, as this variance should be a proxy for heritable variation in size that could be subject to selection.

The opportunity for selection can be interpreted as the extent of variation that could be subject to natural selection and is defined as the ratio of variation in fitness to the square of mean fitness (I = (CV/100)², and [45]. As we consider size at release to be a component of fitness (predicts survival at sea), then in this case we estimate a 20.90% reduction in the opportunity for selection among individuals, and a 35.50% reduction in the opportunity for selection among families (CV among individuals = 7.98 and 8.97, and among families = 3.34 and 4.16 for control and graded, respectively). Thus, the grading treatment was successful in that it did seem to have the desired effect, although whether this would be enough to appreciably slow the rate of domestication selection is unknown at this point.

It is interesting that this reduction in variance was achieved almost entirely by holding back the growth of the initially-large grading group, and of the families that were overrepresented in that group. Relieving the initially-small families from

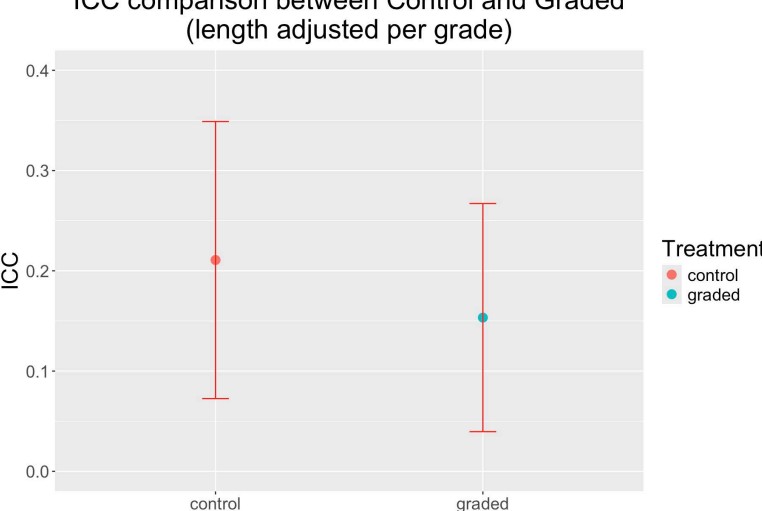

**Fig 6. Intraclass correlation (ICC) values for control and grading treatments.** The intraclass correlation is the fraction of total variance in size that is distributed among families. The broadly overlapping confidence intervals show the ICC was not significantly lower in the grading treatment.

competition with larger fish and raising them in low density tanks with high food availability did little to accelerate their growth relative to the controls. Thus, we conclude that fish in the left-hand tail of the growth distribution are growing slowly due to inherent physiological or behavioral traits that limit their ability to take advantage of the hatchery setting, rather than to suppression by larger, more aggressive fish. Future research focused on what traits characterize those slow-growing fish might point to other environmental modifications one could make to accelerate their growth and further reduce the among-family variation. Until then, reduction in the opportunity for domestication selection that one can achieve by grading will probably be modest.

Grading and discarding small fish is commonly used in hatcheries to achieve more uniformly sized fish at the time of release [46]. Thus, implementing this variation of grading (separating initially-large and initially-small fish, and raising them separately) as a way to reduce variation in size at release might not be too difficult for hatcheries to implement. Here we used feed and density to manipulate growth in the different groups of fish. Fish growth can, of course, be influenced by a range of factors [47–51], so it should be possible for hatcheries to refine their own protocols for slowing down the growth of initially-large juveniles and promoting the growth of initially-small juveniles.

Although grading and separate rearing was an effective way to slow the growth of the initially-large fish, there was little impact on the growth of the initially-smaller fish. While the grading treatment led to some reduction in the opportunity for selection, the average size of the grading group was also smaller than of the control group. As size is correlated to survival at sea [19–21], this would predict fewer returns from the grading group, which would be undesirable in a production hatchery. However, grading might be appropriate in a conservation setting where creating the most wild-like fish possible is the goal, rather than producing the greatest number of returning fish. Grading could be most effective for conservation work if one can find a way to positively influence the growth of the initially-small fish along with reducing growth of the initially-large fish. In this way, hatcheries could maintain fish production while reducing the rate of domestication.

## Supporting information

**S1 File. Program for feed calculations.**
(XLSM)

## Acknowledgments

Thanks to the Oregon Department of Fisheries and Wildlife (ODFW), especially John Spangler and staff at the Siletz Falls trap facility, and to staff at the Oregon Hatchery Research Center and the Aquatic Animal Heath Lab for invaluable assistance in obtaining eggs and raising fish for these experiments. Thanks to Stephanie Bollmann and to OSU's Center for Quantitative Life Sciences for help with genotyping. We also thank Stephanie Bollmann for her guidance in data analysis and interpretation.

## Author contributions

**Conceptualization:** Miriam Obley, Michael S. Blouin.

**Data curation:** Miriam Obley, Stephanie R. Bollmann, Ruth H. Milston-Clements.

**Formal analysis:** Miriam Obley, Stephanie R. Bollmann, Michael S. Blouin.

**Funding acquisition:** Michael S. Blouin.

**Writing – original draft:** Miriam Obley.

**Writing – review & editing:** Michael S. Blouin.

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
