## [Decision Letter · Decision Letter 0]

14 May 2025

*Oncorhynchus mykiss*

Dear Dr. Obley,

Thank you for submitting your manuscript to PLOS ONE. After careful consideration, we feel that it has merit but does not fully meet PLOS ONE’s publication criteria as it currently stands. Therefore, we invite you to submit a revised version of the manuscript that addresses the points raised during the review process.

We look forward to receiving your revised manuscript.

Kind regards,

Abir Ishtiaq, Ph.D.

Academic Editor

PLOS ONE

Journal Requirements:

[This research was supported by the Oregon Hatchery Research Center (https://www.dfw.state.or.us/fish/ohrc/). The funders had no role in study design, data collection and analysis, decision to publish, or preparation of the manuscript.].

3. Please include captions for your Supporting Information files at the end of your manuscript, and update any in-text citations to match accordingly. Please see our Supporting Information guidelines for more information: http://journals.plos.org/plosone/s/supporting-information .

Reviewers' comments:

Reviewer's Responses to Questions

**Comments to the Author**

1. Is the manuscript technically sound, and do the data support the conclusions?

Reviewer #1: Yes

Reviewer #2: Partly

2. Has the statistical analysis been performed appropriately and rigorously?

Reviewer #1: Yes

Reviewer #2: N/A

3. Have the authors made all data underlying the findings in their manuscript fully available?

Reviewer #1: Yes

Reviewer #2: Yes

4. Is the manuscript presented in an intelligible fashion and written in standard English?

Reviewer #1: Yes

Reviewer #2: Yes

Reviewer #1: The manuscript addresses an important issue in hatchery-based fish production, namely the unintended domestication selection that results in reduced fitness of hatchery fish in the wild. The findings provide valuable insight into the biological limits of growth compensation in small fish.

The abstract lacks details on sample sizes and statistical methods used.

The conclusion would benefit from a clearer articulation of the practical implications for hatchery operations.

Please enhance the introduction by including the significance of Oncorhynchus mykiss in aquaculture and human nutrition. For instance, O. mykiss is recognized as a nutritionally valuable fish species with high protein content (Naeem et al., 2016). This will help establish the biological and economic importance of the species under study.

Please include the geographical coordinates of the study area for reproducibility and clarity.

Specify the dose of MS-222 (tricaine methanesulfonate) used for euthanizing the fish to ensure compliance with animal welfare standards.

Review the numbering of the figure titled "Distribution of families throughout the grading tanks." It appears it should be labeled as Figure 4 instead of the current designation.

The discussion section would benefit from more comprehensive analysis. Specifically, include various biological and environmental factors that may influence fish growth patterns, such as habitat, sex, and feeding regimes. For example, fish growth may be influenced by a range of factors (Javed et al., 1992; Naeem et al., 2010; Naeem et al., 2011; Yousaf et al., 2011; Ismat et al., 2013). Integrating such insights will enhance the depth and applicability of your conclusions.

Ensure that all cited references are arranged chronologically in the text (e.g., Lines 75, 76, 87–88, 356).

Revise all in-text citations to conform to the journal’s prescribed formatting style (e.g., Lines 78, 89–90, 95, 330).

Add the publication year for "Li and Brocksen, n.d." if available or clarify it as unpublished data or personal communication as appropriate.

Consider incorporating more recent references to strengthen the study's relevance and contextual background.

Reviewer #2: It seems that the experiment has to independent factors, which are hatchery conditions (grading and feeding regime) and the stocking density in the large-sized tank. The authors ignore the impact of higher biomass in the treated group, which definitely affects the performance of the tested fish.

all comments are included in the manuscript file

**Do you want your identity to be public for this peer review?** For information about this choice, including consent withdrawal, please see our Privacy Policy

Reviewer #1: **Yes: ** Prof. Dr. Muhammad Naeem

Reviewer #2: No

---

## [Author Response · Author response to Decision Letter 1]

3 Nov 2025

Associate Editor, Review 1, and Review 2,

Thank you for your valuable and insightful feedback. The comments have been addressed in the manuscript and in responses below.

Reviewer #1: The manuscript addresses an important issue in hatchery-based fish production, namely the unintended domestication selection that results in reduced fitness of hatchery fish in the wild. The findings provide valuable insight into the biological limits of growth compensation in small fish.

The abstract lacks details on sample sizes and statistical methods used.

Added

The conclusion would benefit from a clearer articulation of the practical implications for hatchery operations.

Added

Please enhance the introduction by including the significance of Oncorhynchus mykiss in aquaculture and human nutrition. For instance, O. mykiss is recognized as a nutritionally valuable fish species with high protein content (Naeem et al., 2016). This will help establish the biological and economic importance of the species under study.

Added

Please include the geographical coordinates of the study area for reproducibility and clarity.

Specify the dose of MS-222 (tricaine methanesulfonate) used for euthanizing the fish to ensure compliance with animal welfare standards.

Added

Review the numbering of the figure titled "Distribution of families throughout the grading tanks." It appears it should be labeled as Figure 4 instead of the current designation.

Corrected

The discussion section would benefit from more comprehensive analysis. Specifically, include various biological and environmental factors that may influence fish growth patterns, such as habitat, sex, and feeding regimes. For example, fish growth may be influenced by a range of factors (Javed et al., 1992; Naeem et al., 2010; Naeem et al., 2011; Yousaf et al., 2011; Ismat et al., 2013). Integrating such insights will enhance the depth and applicability of your conclusions.

Added

Ensure that all cited references are arranged chronologically in the text (e.g., Lines 75, 76, 87–88, 356).

Corrected

Revise all in-text citations to conform to the journal’s prescribed formatting style (e.g., Lines 78, 89–90, 95, 330).

Add the publication year for "Li and Brocksen, n.d." if available or clarify it as unpublished data or personal communication as appropriate.

Corrected

Consider incorporating more recent references to strengthen the study's relevance and contextual background.

Added

Reviewer #2: It seems that the experiment has to independent factors, which are hatchery conditions (grading and feeding regime) and the stocking density in the large-sized tank. The authors ignore the impact of higher biomass in the treated group, which definitely affects the performance of the tested fish.

We are not testing the main effects of feed and density, which are well known to influence growth and are not interesting. These are simply the tools we used to manipulate the subsequent growth of each initial-size groups (S, M & L). What we are comparing is the variance in size among fish in the final graded cohort (S,M & L together) versus the variance in the control (ungraded) cohort. From the reviewer’s comment, it appears we did not adequately explain the goals of the study. Therefore, we have added a new figure in the introduction that we hope will clarify the whole point of the study.

all comments are included in the manuscript file

6. PLOS authors have the option to publish the peer review history of their article (what does this mean?). If published, this will include your full peer review and any attached files.

Do you want your identity to be public for this peer review? For information about this choice, including consent withdrawal, please see our Privacy Policy.

Reviewer #1: Yes: Prof. Dr. Muhammad Naeem

Reviewer #2: No

In response to the last comment by Rev #2 that was embedded in the pdf:

We are confused by this comment about the discussion: “the discussion is very poor. lacks in comparison with the previous work. rewrite it and concern both rearing conditions and genetic impact.”

Firstly, to our knowledge, no one has ever done this type of study. The goal was to reduce to total variance in size at release, while keeping all the fish, as a means to reduce the opportunity for domestication selection. Grading has been used in the past to achieve uniform size by throwing away the initially-small fish. We are unaware of other data on the variance in size at release in graded vs ungraded cohorts in which all fish were retained. So, we have no other literature to compare our work to. If there is a relevant reference we missed, we would be grateful to have it pointed out to us.

To be clear, what this paper isn’t about is the effects of crowding or food levels on growth rate. Those are just tools we used to manipulate the subsequent growth rates of the initially-large and initially-small groups of juveniles. To help readers understand the point of the work, we now have added a figure in the introduction that illustrates the theoretical premise of the work and what successful results of the manipulation would look like.

Secondly, our discussion is entirely in the context of rearing conditions and genetics. We discuss the opportunity for selection, which is what we are trying to reduce (the ‘opportunity for selection’ is a standard term in quantitative genetics). Therefore, we apologize, but we aren’t sure what exactly we are supposed to change about the discussion. The results and their interpretation in terms of the potential for domestication selection seem very straightforward to us, and we discussed them as such. We did add the references Dr. Naeem suggested. However, what else we are requested to change is not clear to us.

Respectfully,

Miriam Obley

---

## [Decision Letter · Decision Letter 1]

25 Nov 2025

Grading by size to reduce the opportunity for domestication selection in hatchery-reared steelhead (*Oncorhynchus mykiss* )

PONE-D-25-01027R1

Dear Dr. Miriam Obley,

We’re pleased to inform you that your manuscript has been judged scientifically suitable for publication and will be formally accepted for publication once it meets all outstanding technical requirements.

Kind regards,

Silvia Martínez-Llorens

Academic Editor

PLOS ONE

Reviewers' comments:

Reviewer's Responses to Questions

**Comments to the Author**

Reviewer #1: All comments have been addressed

Reviewer #2: All comments have been addressed

2. Is the manuscript technically sound, and do the data support the conclusions?

Reviewer #1: Yes

Reviewer #2: Yes

3. Has the statistical analysis been performed appropriately and rigorously?

Reviewer #1: Yes

Reviewer #2: N/A

4. Have the authors made all data underlying the findings in their manuscript fully available?

Reviewer #1: Yes

Reviewer #2: Yes

5. Is the manuscript presented in an intelligible fashion and written in standard English?

Reviewer #1: Yes

Reviewer #2: Yes

Reviewer #1: Authors have adequately addressed suggested comments raised in a previous round of review and manuscript is technically sound hence I feel that this manuscript is now acceptable for publication.

Reviewer #2: (No Response)

**Do you want your identity to be public for this peer review?** For information about this choice, including consent withdrawal, please see our Privacy Policy

Reviewer #1: No

Reviewer #2: **Yes: ** Hadir A. Aly

---

## [Editor Report · Acceptance letter]

PONE-D-25-01027R1

PLOS One

Dear Dr. Obley,

I'm pleased to inform you that your manuscript has been deemed suitable for publication in PLOS One. Congratulations! Your manuscript is now being handed over to our production team.

Kind regards,

on behalf of

Dr Silvia Martínez-Llorens

Academic Editor

PLOS One